# Smart Systems for Material and Process Designing in Direct Nanoimprint Lithography Using Hybrid Deep Learning

**DOI:** 10.3390/nano12152571

**Published:** 2022-07-27

**Authors:** Yoshihiko Hirai, Sou Tsukamoto, Hidekatsu Tanabe, Kai Kameyama, Hiroaki Kawata, Masaaki Yasuda

**Affiliations:** 1Physics and Electronics Engineering, Graduate School of Engineering, Osaka Metropolitan University, Sakai 588-8531, Japan; sab01044@st.osakafu-u.ac.jp (K.K.); kawata_hiroaki@omu.ac.jp (H.K.); yasuda.masaaki@omu.ac.jp (M.Y.); 2Physics and Electronics Engineering, Graduate School of Engineering, Osaka Prefecture University, Sakai 599-8531, Japan; sougen0530@icloud.com (S.T.); tana0110hide@icloud.com (H.T.)

**Keywords:** direct nanoimprint, process design, deep learning

## Abstract

A hybrid smart process and material design system for nanoimprinting is proposed, which is combined with a learning system based on experimental and numerical simulation results. Instead of carrying out extensive learning experiments for various conditions, the simulation learning results are partially complimented when the results can theoretically be predicted by numerical simulation. In other words, the data that are lacking in experimental learning are complimented by simulation-based learning results. Therefore, the prediction of nanoimprint results without experimental learning could be realized under various conditions, even for unknown materials. In this study, material and process designs are demonstrated for a low-temperature nanoimprint process using glycerol-containing polyvinyl alcohol. The experimental results under limited conditions were learned to investigate the optimum glycerol concentrations and process temperatures. Simulation-based learning was used to predict the dependence on press pressure and shape parameters. The prediction results for unknown glycerol concentrations agreed well with the follow-up experiments.

## 1. Introduction

In conventional photo lithography, nanostructures are obtained by processing photosensitive materials. On the other hand, nanoimprint lithography [1,2] directly achieves nanofabrication on various materials, for example, organic semiconductors [3,4,5] and low-melting-point glass [6,7,8]. This enables various applications in electronics, photonics and biodevices, which have been difficult to realize using conventional lithography. Both optimum process design and defect elimination are common issues. In conventional lithography, computational lithography has significantly improved for process design. Attempts have also been made to use artificial intelligence to predict defect locations (hotspots) and optimize layout. A successful reason for this is that the use of highly standardized materials and equipment make it suitable for deep learning systems based on huge databases and optimizing pattern layouts that are realized to eliminate defects [9,10,11,12].

On the other hand, the process conditions are determined by mechanical properties such as the viscoelastic modulus of the material and topological conditions such as pattern width, pattern height and pressure [13,14,15,16] in nanoimprint lithography. Therefore, the process conditions are determined by simulation analysis and empirical knowledge based on the measurements of mechanical properties and topological conditions [17]. One feature of nanoimprinting is the use of application-specific materials. This requires the optimization of the process, including pattern layout, according to the material’s properties.

However, for materials with unknown properties or new processes, it is necessary to carry out a series of preliminary experiments, measure the material properties and accumulate nanoimprinting results, which means that process and material optimization takes an enormous amount of time.

The aim of this study was to propose a smart manufacturing system that provides an optimized process for a wide variety of materials and structures using an artificial neural network system [18,19,20] that can handle novel processes and materials based on a knowledge base of experiments and simulations.

## 2. Low-Temperature Direct Nanoimprint for Polyvinyl Alcohol (PVA) with Glycerol Additives

The aim of this study was to develop a low-temperature direct nanoimprint process for PVA, which is a water-soluble polymer and has great value for its use as a disposable mold and sacrificial layer. However, the water solubility of PVA is impaired when heated above about 150 °C. Therefore, the process temperature should be below 150 °C. However, higher pressing pressure is needed, and this causes defects in the process.

To solve this problem, we found that the pressing temperature of PVA can be lowered by adding glycerol to PVA. Figure 1 demonstrates the direct nanoimprint to glycerol-added PVA. The effect of glycerol additive is evidently revealed.

However, there is no easy way of finding the optimal additive concentration of glycerol, and many investigation experiments are required to confirm the optimal conditions by successively changing the amount of glycerol that is added and the process temperature. On the other hand, for the process simulation, the material parameters, such as elastic modulus for thin film, should be measured.

Therefore, we attempted to construct an optimization system for nanoimprint materials and their processes using artificial intelligence.

## 3. Hybrid Deep-Learning System for Nanoimprint Material and Process

To realize a smart design system for nanoimprint materials and processes, we propose a novel system to predict the pattern height after thermal nanoimprinting by learning based on experimental data and numerical simulation data, where the simulation learning complements the learning results which could not be carried out in experiments. We refer to this system as a hybrid system, which combines learning from experiments with learning from numerical simulations. Figure 2 shows the system diagram. The input parameters are process conditions such as pressure and temperature, material characteristics such as solvent concentration, and the topological structure such as the pattern size and feature for each learning system. In this case, the output is the pattern height after nanoimprinting.

The hidden layer is fixed to be two layers and the numbers of each node *k*_1_ and *k*_2_ are set to obtain less evaluation error in the learning process, as shown in Figure 3.

In experimental learning, the learning data were extracted by varying the process conditions by variations in the Glycerol concentration and process temperature. Therefore, topological conditions such as mold groove width or depth and process conditions such as pressing pressure were fixed to minimalize the number of experiments.

In the simulation learning, simulation results under some representative conditions were used for learning, where the data were extracted using polymethyl methacrylate (PMMA) as a reference material instead of using actual material. The importance of this is that the nanoimprinted results were reported in our simulation and experimental studies [13,14,15], where the results were determined by the relative pressure on the modulus of the material and the topology of the mold based on the similarity laws of the continuum mechanics.

The simulation learning could calculate dependencies on pattern shape, press temperature, press pressure and the elastic modulus of the material, but could not handle dependencies on unknown materials for which the physical parameters were unknown. As the system is a normalization system based on continuum mechanics. Under the law of similarity, the relative value of the normalized pattern height can be derived from the normalized pressure P/E, where P is the press pressure and E is the modulus of the material.

In other words, if the mechanical properties of the polymers are relatively similar, the normalized relative press pressure can be obtained if the normalized pattern height is known. As a result, the relative pressure can be estimated even if the elastic modulus of the material is unknown.

Furthermore, if the relative press pressure is known, the pattern height after nanoimprinting can be predicted by changes in the press pressure.

Thus, under certain approximations, simulation can complement the learning results.

In this study, PVA with glycerol was considered a rubber elastic material and the Mooney–Rivlin model was used to represent the mechanics, with the Mooney coefficient determined using a well-known empirical formula [21,22].

From these learning results, the optimum concentration of glycerol could be designed and process conditions could be proposed to obtain the required pattern.

In the experimental learning, the dependence of pattern height on glycerol concentration and press temperature is predicted. However, the dependence on press pressure could not be predicted because the learning data were not taken into account.

The elastic modulus of PVA with glycerol was substituted for that of polymethyl methacrylate (PMMA). According to the similarity rule, it was assumed that the pattern shape after nanoimprinting is uniquely determined by the relative values P/E of the pressing pressure P and the elastic modulus E, as described above.

Thanks to this approximation, the measurement of the actual elastic modulus of each polymer is omitted and the relative relationship of the pattern height to the press pressure can be obtained. The same can be applied to changes in mold topography.

For deep learning, a multi-layer back-propagation neural network (BPNN) system, which is one of the most representative deep learning systems among machine learning systems, is used to predict the nanoimprinting results and propose nanoimprinting processes and materials. The activation function was the conventional Sigmoid function. The system is a homemade system using Python [23] convened with Microsoft Excel [24].

The number of nodes in hidden layers are determined by the evaluation results of the sum of squared error in testable learning. Figure 3 shows typical results using experimental learning data. In this case, *k*_1_ = 10 and *k*_2_ = 10 give less than 0.05 in sum of square errors. Based on the evaluation, both learning systems consist of two hidden layers of 10 nodes that consider the prediction error. The number of learning data were 43 and 120, for experiments and simulation data, respectively, and 70–85% of them were used for training. We use a simple in-house optimizer based on gradient descent method and the Numpy utilities (https://numpy.org/, accessed on June 2020). The typical learning coefficient was set to 0.01 and the number of learning epochs was approximately 5000 or more, which takes several hours using conventional personal computers without a normal Graphics Processing Unit (GPU).

## 4. Results and Discussions

### 4.1. Characterization of PVA Containing Glycerol and Prediction of Pattern Formability

First, the pattern formability of glycerol-containing PVA was experimentally investigated. Here, the nanoimprinted result is simply characterized as the pattern height after nanoimprinting.

Figure 4a shows our previous experimental results [20] of the cross-section images of the pattern after nanoimprinting. In this experiment, the weight concentration of glycerol was varied from 0 to 25%. The pressing temperatures were 100, 130, and 150 °C. The press temperatures were 100 °C, 130 °C and 150 °C. The press pressure was 10 MPa, the line width of the pattern was 2 µm, the groove depth of the mold was 2 µm, and the thickness of the PVA was 3 µm, respectively.

The nanoimprinting was performed at least three times under each condition, for a total of 43 experiments. These were used as learning data.

The diamonds, triangles, and squares in Figure 4b show the experimental results of the average height of the patterns under each condition as a learning data. It is difficult to visually extract the regularity from the graph.

In the learning process, we determined the weight function of the neural networks as described in Figure 3 for an experimental learning system. Approximately 70% of the experimental data were used as training data to learn the relationship between glycerol concentration, process temperature, and pattern height after nanoimprinting.

Solid lines in Figure 4b show the prediction results by deep learning. At various temperatures, the pattern height has a local peck at around 10% of the glycerol concentration. It can be predicted that a 10% glycerol concentration and a press temperature that is as low as possible—around 130 °C—will more effectively improve pattern formability. It looks fairly good to express the experimental results.

Based on the learning results (weight function), we predicted the dependence of the glycerol concentration on varying process temperatures.

Figure 5 shows the results of the predicted pattern height at temperatures not present in the learning experimental data, with solid lines for 140 °C and 120 °C.

The dots in the figure show the additional experimental results for validation.

The predictions overestimated the validation experiments.

The reason for this is the complicated behavior of the learning data in the lower glycerol concentration below 10%. The pattern height at 130 °C and 150 °C in the learning data (Figure 4b) appears to be largely inverted below 10% of glycerol concentrations. Apart from the accuracy of the experimental learning data and its physical reasons, this large inversion which could cause a prediction error at 140 °C, just halfway between 130 °C and 150 °C, especially in the region of glycerol concentrations below 10%.

On the other hand, for 120 °C, between 130 °C and 110 °C, the predicted results are almost a good representation of the experiment, even when the glycerol concentration is below 10%.

To improve the accuracy, more training data are required for several temperature ranges when the glycerol concentration is below 10%.

However, compared to the conventional experimental method of conducting experiments under all conditions, this method is firstly useful to obtain a fast outlook for solutions.

Based on deep learning, the pattern height after nanoimprinting could be predicted with a constant pressure, arbitrary temperature, and glycerol concentration.

However, the pressure dependence was not included because it was not considered in the learning experiments.

### 4.2. Pattern Height Prediction by Simulation-Based Learning

To predict the pattern shape after nanoimprinting in relation to the mold shape, polymer topography and press pressure, a learning system was constructed to predict the pattern height using deep learning from representative shape conditions and relative pressure conditions, learned in advance by simulation.

First, a comparison was made between the simulation and experiment. Here, the deformation state of PMMA was compared using a rubber elastic body model. The Young’s modulus E of PMMA at the pressing temperature was measured beforehand, and the results of the experiment and simulation are shown when PMMA was pressed at a relative pressure P/E to the pressing pressure P. Although the two are not in perfect agreement, they are in relative general agreement, and the simulation confirms that the formability can be predicted to some extent.

Then, to prepare the learning data, the relationship between the aspect ratio of the mold grooves and the height of the pattern when the relative pressing pressure was varied was investigated by simulation.

Figure 6a shows some examples of the results. Here, the initial polymer film thickness was set to three times the mold groove depth, and the aspect ratio (groove depth/width) of the mold groove structure was varied.

Figure 6b shows the prediction results of the pattern height after nanoimprinting for mold patterns in various aspect ratios, with varying levels of applied pressure, by deep learning using simulation results as learning data. Due to the prediction accuracy, the results are partly crossed in detail, but this is not a fatal issue with regard to the objective of predicting rough results. What is important is that the system determines the weight function in the neural networks, and then the deep learning provides the pattern height after nanoimprinting under arbitrary process conditions without a fudge computational cost of numerical simulation.

### 4.3. Hybrid System for Material and Process Design for Nanoimprinting

We demonstrated a system for learning pattern heights with a varying glycerol concentration in PVA and process temperature with a fixed press pressure and mold topography, and a system for learning and predicting nanoimprinting results for variations in press pressure and pattern topography, complementing the simulation-based learning.

The former system uses learning data based on experiments, while the latter system uses simulation results using PMMA as a representative reference polymer as learning data.

As the results of nanoimprinting are mostly determined by the mechanical property of the polymer, i.e., the elastic modulus or viscoelasticity of the polymer, according to the law of similarity, the PVA containing glycerol behaves in a similar way to those of reference polymers, such as PMMA, with an equivalent relative modulus of the polymer.

In other words, if the relative press pressure is equivalent to the modulus of elasticity of the polymer, the nanoimprinting results will be the same.

The procedure of the prediction is shown in Figure 7 for glycerol-containing PVA.

First, we chose the glycerol concentration, referring to Figure 5. For instance, when the press temperature was set to 120 °C, we chose 10% to be the pattern height with a local peak (A in Figure 7a).

The pattern height was predicted to be approximately 1.4 µm. This height corresponds to a relative height of approximately 0.6, where the mold depth is 2.3 µm.

Then, we predicted pattern height for various nanoimprinting pressures at 120 °C The learning results from the simulation were used to predict the relationship between the pressure and nanoimprinting height for the relative aspect ratio of 1.15, as shown in Figure 7b. The relative press pressure (P/E) for a relative height of 0.6 can be read from the graph as approximately 0.65 (B in Figure 7b); namely, the experimental pressing pressure of 10 MPa for PVA could be regarded as pressing at a relative pressure of 0.65.

Therefore, the nanoimprinting result, when pressed at, for example, 20 MPa, can be traced at a relative pressure of 1.3 (C in Figure 7c), which is twice the relative pressure at 10 MPa.

This operation can be automatically charted by a macro-program such as Excel, based on the results of deep learning predictions.

As such, the relationship between the actual pressing pressure and the nanoimprinting height can be linked and predicted under mostly material, process, and topology conditions.

As a result, experimental results with missing pressure and geometry conditions could be complemented by learning simulation results based on similarity rules to predict nanoimprint results. This minimizes the amounts of learning data needed for unknown materials with many parameters to be varied.

Figure 8 shows the predicted results and follow-up experimental results for various pressures at a nanoimprint temperature of 120 °C and 140 °C and glycerol concentration of 10%. The solid line shows the pattern height predicted by the hybrid learning, and the square and triangle show the experimental results at 5 MPa, 10 MPa, and 15 MPa, respectively. The deep learning predictions are in fairly good agreement with the experimental results. However, the prediction overestimated as well as Figure 5, especially for 140 °C. However, the prediction overestimated especially for 140 °C. The reason is the complex behavior of the learning data, as discussed in Figure 5. However, if we need much more learning data, however, the general trends and process design guidelines are obtained, and are sufficient as a guideline for process design. On the other hand, collecting more training data is as time-consuming and labor-intensive as conventional optimization methods that rely on experimentation. Therefore, the results obtained are well worth the effort.

In summary, we proposed a system that combines deep learning from experiments to characterize material properties and deep learning from simulation results to predict formability under varying process conditions. It is shown that these systems can complement each other’s learning results, enabling a rough prediction of nanoimprinting performance for unknown materials under unknown process conditions.

According to this approach, it is possible to predict optimal process and material conditions by learning multiple nanoimprint results for an unknown material, even without knowing the mechanical properties of the material that are important for nanoimprinting, namely the elastic modulus. This could be useful for the direct nanoimprint process design.

The remaining issue will be to verify the extent to which the similarity rule—which, in this case, treated the well-known PMMA as a standard material—can be adapted to other resin materials and functional materials.

## 5. Conclusions

We proposed a smart design system for nanoimprinting processes and materials based on a hybrid learning system from experiments and numerical simulation results.

We believe that this smart system can provide a good prediction of results for unknown materials in direct nanoimprinting with diverse materials.

In the future, the system will be extended to various materials and topological structures to confirm the results.

## Figures and Tables

**Figure 1 nanomaterials-12-02571-f001:**
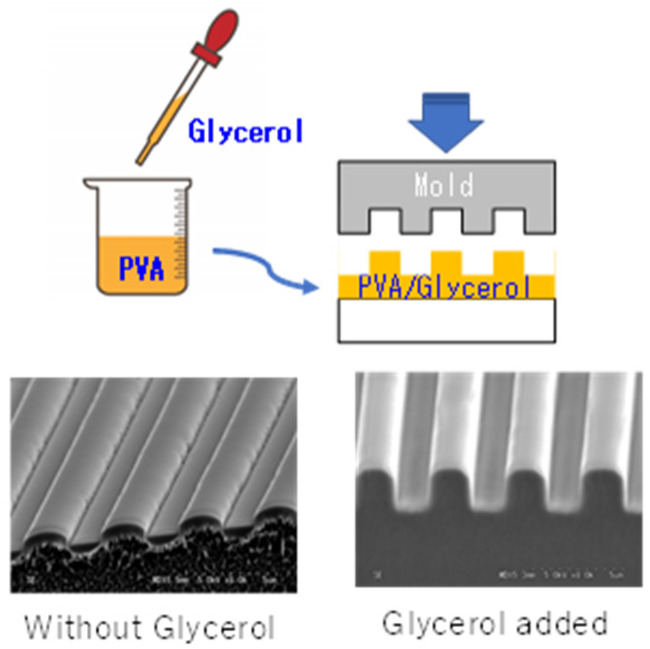
Direct nanoimprint results with and without glycerol additives to PVA (10 MPa, 130 °C, Line width = 2.0 μm).

**Figure 2 nanomaterials-12-02571-f002:**
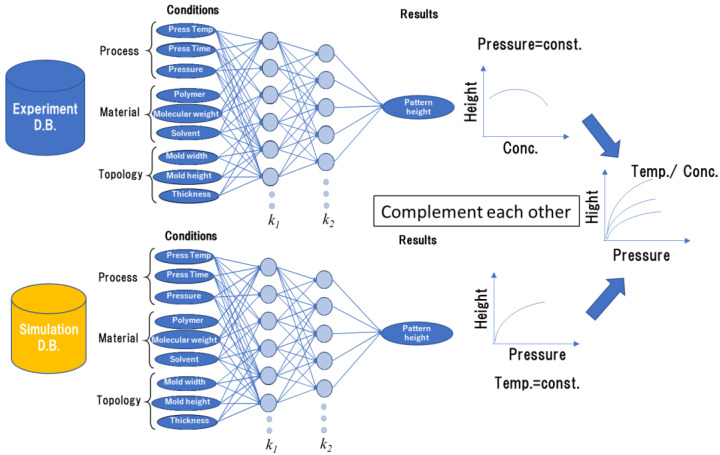
Schematics of a ‘hybrid’ deep learning system in combination with experiment and simulation.

**Figure 3 nanomaterials-12-02571-f003:**
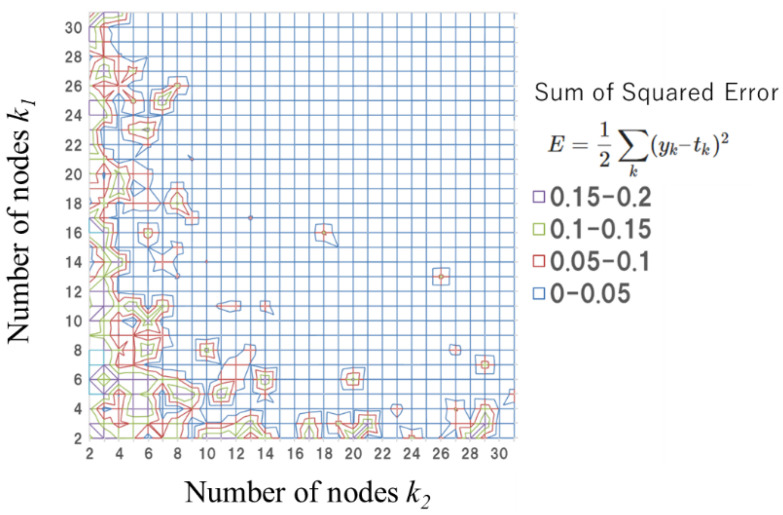
Example of accuracy evaluation map in sum of square error for various numbers of nodes in hidden layers based on experimental learning data based on the results shown in Figure 3.

**Figure 4 nanomaterials-12-02571-f004:**
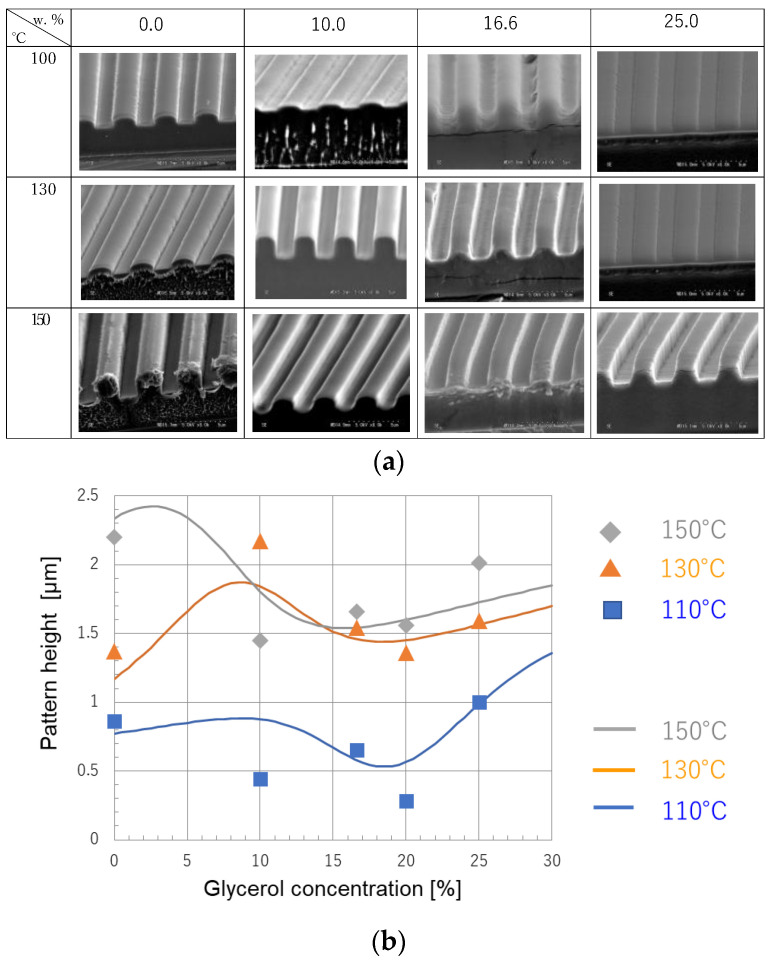
Experimental learning data and learning results. (**a**) The scanning electron microscope (SEM) images of experimental results for PVA with glycerol (pressure: 10 MPa, 2.0 μm line and space). (**b**) Experimental results for learning data at various glycerol concentrations under limited process conditions. Diamonds, triangles, and squares are averages of the experimental results. Solid lines are the predicted results by learning for various glycerol concentrations.

**Figure 5 nanomaterials-12-02571-f005:**
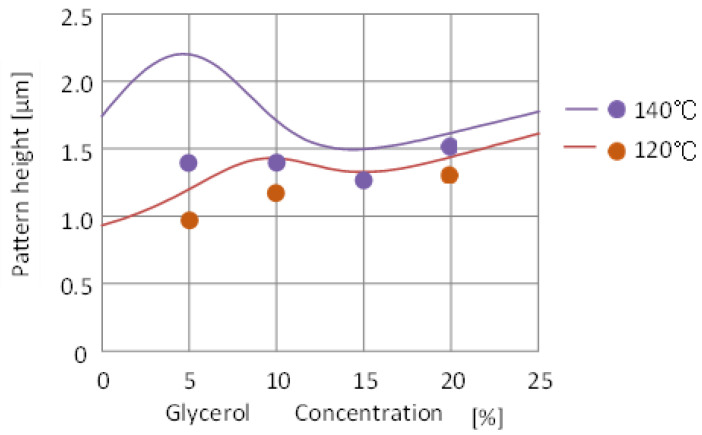
Predicted pattern height (solid lines) and verification experimental results (dots) for various glycerol concentrations at 120 °C and 140 °C based on experimental learning data at 110 °C, 130 °C, and 150 °C.

**Figure 6 nanomaterials-12-02571-f006:**
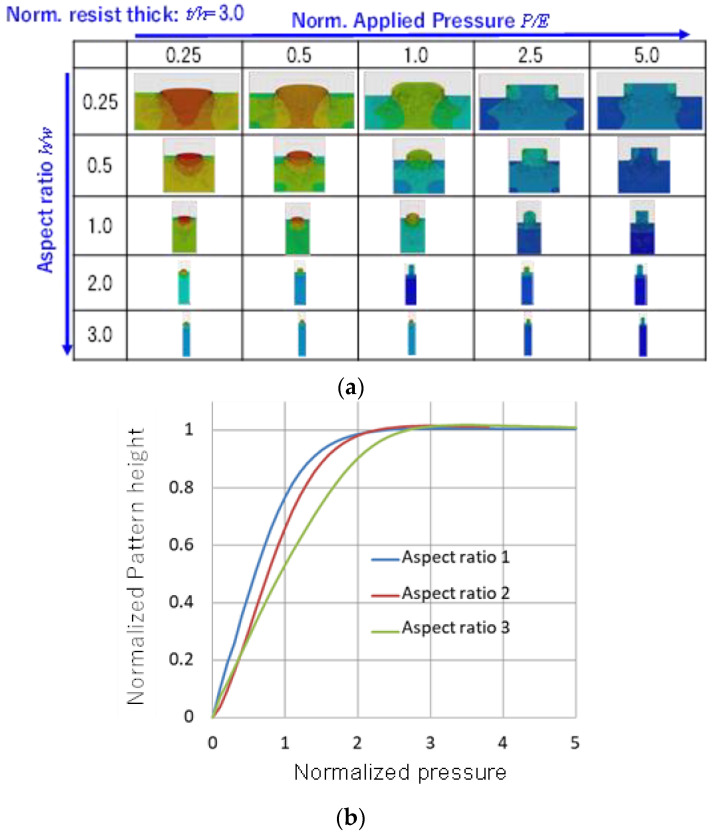
Learning data by numerical simulation for a compliment of pressure and geometrical structural dependence. (**a**) Simulation results for learning data; and (**b**) Prediction results of the pattern height after nanoimprinting for mold patterns with various aspect ratios, varying with applied pressure using deep learning.

**Figure 7 nanomaterials-12-02571-f007:**
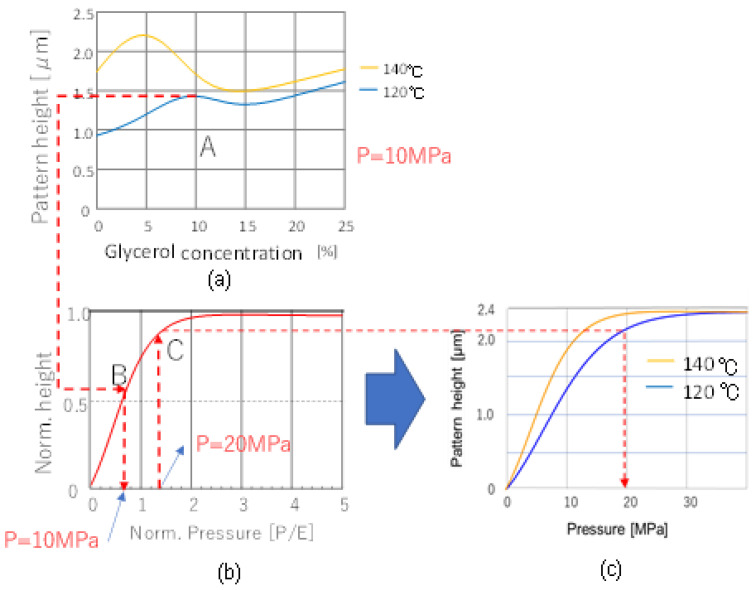
Diagram of the procedure for prediction by hybrid learning system. (**a**) Decide Glycerol concentration for example 10% and check pattern height, (**b**) Look normalized pattern height and find corresponded normalized pressure (in this case, it is equivalent to 10MPa), (**c**) Draw transformed relation between applied pressure versus pattern height after nanoimprinting based on (**b**).

**Figure 8 nanomaterials-12-02571-f008:**
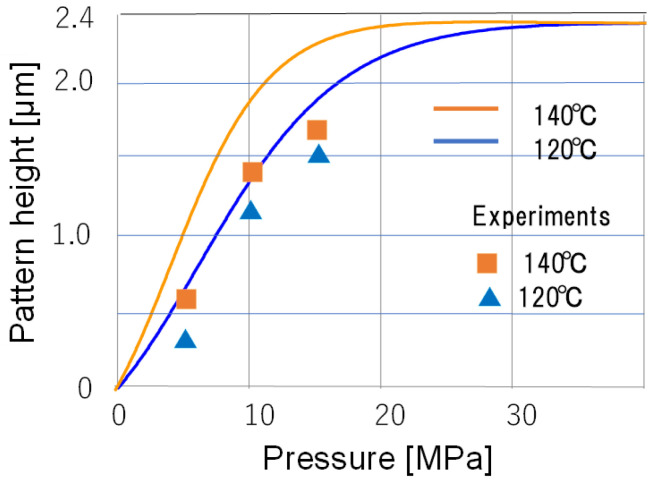
Prediction of pattern height with varying pressure by hybrid deep learning and experimental results. The nanoimprinting temperature was 120 °C and 140 °C and the glycerol concentration was 10%.

## Data Availability

Sou Tsukamoto, Graduation thesis of Osaka Prefecture University, College of Engineering (2021, March) *in Japanese* (Not open access).

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
