# Peer review of "Smart Systems for Material and Process Designing in Direct Nanoimprint Lithography Using Hybrid Deep Learning"

_nanomaterials, 2022, doi:10.3390/nano12152571_

Round 1
Reviewer 1 Report
The authors proposed a hybrid smart systems for material and process designing in direct nanoimprint lithography using hybrid deep learning. Before publication, the authors should address following issue as a major revision.
1. Better to add some comments for which sample is desired in the upper figure in Figure 3. and better to distinguish upper and lower figures as (a), (b), which is same with Figure 6.
2. Experimetnal results and expected value is quite different in Fig. 4. Better to add some discussion for the reason.
Author Response
Thank you for your delightful suggestions.
Firstly, I apologize our fatal mistakes about Fig. 8 of their prediction graphs.
Figure 8 is replaced to correct one in revised manuscript.
Also, additional description is added in line 298-300 as :
However, the prediction overestimated as well as Fig.5, especially for 140°C. However, the prediction overestimated especially for 140°C. The reason is complex behavior of the learning data as discussed in Fig.5.
- Better to add some comments for which sample is desired in the upper figure in Figure 3. and better to distinguish upper and lower figures as (a), (b), which is same with Figure 6.
Yes, we revised the figures as Figure 4 and 6, respectively.
Figure 3 is newly added in revised manuscript. 
Figures 3 and 5 in original manuscript are merged to Figure 4(b) in revised manuscript
- Experimental results and expected value is quite different in Fig. 4. Better to add some discussion for the reason.
To discuss the reason, Fig.3 (lower) , which shows experimental results and Fig.5 (prediction results) are merged to Fig. 4 (b).
We revised and added discussions about Fig.5 in line 194-207 as :
The predictions overestimated the validation experiments.
The reason for this is complicated behavior of the learning data in lower Glycerol con-centration below 10%. The pattern height at 130°C and 150°C in the learning data (Fig. 4(b) ) shows largely inverted below 10% of Glycerol concentrations. Apart from the accuracy of the experimental learning data and its physical reasons, due to this large inversion could cause prediction error at 140°C, just halfway between 130°C and 150°C, especially in the region of glycerol concentrations below 10%.
On the other hand, for 120 °C, between 130 °C and 110 °C, the predicted results are almost a good representation of the experiment, even when the glycerol concentration is below 10%.
To improve the accuracy, more training data is required for several temperature ranges when the glycerol concentration is below 10%.
However, compared to the conventional experimental method of conducting experiments under all conditions, this method is firstly useful to get fast outlook for solutions.
I'll take you the time and effort but Please review again.
Thank you.
Yoshihiko HIRAI

Reviewer 2 Report
Dear Authors,
For reproducibility sake, you need to give more details about the Artificial Neural Networks you use:
1. Having a diagram showing the nets is good (from your article I can see that you use a feed-forward network with 4 layers: the input layer contains 9 input neurons (the input is a vector of dimension 9), the first hidden layer contains 6 neurons, the second hidden layer contains 5 neurons, the output layer only one neuron). Is that really the case?
2. Which activation functions did you use for the hidden layers? From my understanding of the paper you are using the network to predict a continuous value (i.e., you are doing a regression). What is the activation function for the output neuron (is it linear?)?
3. How did you prepare your data. Are the input normalized? And if so, how?
4. How did you train the network (which optimizer are you using? how did your split the data between training, validation and test sets? how many epochs? what is the batch size? etc.)
5. How did you evaluate the goodness of fit of the network? How do you know that the network is not too complex and is overfitting your data? Do you think another type of data-driven model (like a SVM, or a polynomial regression) would work as well?
6. How did you implement the network: which library are you using. Are you planning to open source your code?
I am absolutely not an expert in the field of direct nanoimprint lithography so I cannot give any feedback on that part, but as an expert in the field of data science, I can say that the part describing the machine learning setup you are using is not detailed enough for anyone else to reproduce it or even judge whether your approach is sound.
Author Response
Thank you for fine suggestions and comments.
Firstly, I apologize our fatal mistakes about Fig. 8 of their prediction graphs.
Figure 8 is replaced to correct one in revised manuscript.
Also, additional description is added in line 298-300 as :
However, the prediction overestimated as well as Fig.5, especially for 140°C. However, the prediction overestimated especially for 140°C. The reason is complex behavior of the learning data as discussed in Fig.5.
The followings are your suggestions and our reply.
- Having a diagram showing the nets is good (from your article I can see that you use a feed-forward network with 4 layers: the input layer contains 9 input neurons (the input is a vector of dimension 9), the first hidden layer contains 6 neurons, the second hidden layer contains 5neurons, the output layer only one neuron). Is that really the case?
In original manuscript on line 133, where we described the number of nodes (neurons).
However, it is not sufficient description. So, Fig.2 is revised where k1 and k2 are number of nodes for each layer. The number of hidden layers was fixed to be 2, and the number of nodes were set to obtain less learning errors.
So, we add description on revised manuscript at lines 84-85 and136-138 as:
The hidden layer is fixed to be 2 layers and the number of each node k1 and k2 are set to obtain less evaluation error in learning process as shown in Fig.3.
The number of nodes in hidden layers are determined by the evaluation results of the sum of squared error in testable learning. Figure 3 shows typical results using experimental learning data. In this case, k1=10 and k2=10 give less than 0.05 in sum of square errors. Based on the evaluation, both learning systems consist of two hidden layers of 10 nodes that consider the prediction error. The number of learning data were 43 and 120, for experiments and simulation data, respectively and 70-85% of them were used for training.
- Which activation functions did you use for the hidden layers? From my understanding of the paper you are using the network to predict a continuous value (i.e., you are doing a regression).What is the activation function for the output neuron (is it linear?)?
We used conventional simple Sigmoid function, added on line 134 for revised manuscript as:
The activation function was conventional Sigmoid function.
- How did you prepare your data. Are the input normalized? And if so, how?
I’m sorry I could not well catch your question but the learning data (pattern height in experiments) were normalized in the system (divided by the mold height (depth)).
For simulated data as well.
Other input data were normalized in the system in automatically.
- How did you train the network (which optimizer are you using? how did your split the data between training, validation and test sets? how many epochs? what is the batch size? etc.)
We briefly described on original manuscript on line 129-135 as:
For deep learning, a multi-layer back-propagation neural network (BPNN) system, which is one of the most representative deep learning systems among machine learn-ing systems, is used to predict the nanoimprinting results and propose nanoimprinting processes and materials. The system is a homemade system using Python [23] con-vened with Microsoft Excel [24]. Both learning systems consist of two hidden layers of 11 nodes that consider the prediction error. The typical learning coefficient was set to 0.01 and the number of learning epochs was around 5,000 or more, which takes several hours using conventional personal computers without a normal Graphics Processing Unit (GPU).
In addition, we added other information on revised manuscript on line 142-143.
We use simple in-house optimizer based on Gradient descent method and the Numpy utilities (https://numpy.org/).
- How did you evaluate the goodness of fit of the network? How do you know that the network is not too complex and is overfitting your data? Do you think another type of data-driven model (likea SVM, or a polynomial regression) would work as well?
Yes, we newly added typical error evaluation results as Fig. 3 and description on line 136-138 as:
The number of nodes in hidden layers are determined by the evaluation results of the sum of squared error in testable learning. Figure 3 shows typical results using experimental learning data. In this case, k1=10 and k2=10 give less than 0.05 in sum of square errors.
.
But currently we do not know how to evaluate the network or upgrade the model.
- How did you implement the network: which library are you using. Are you planning to opensource your code?
I am absolutely not an expert in the field of direct nanoimprint lithography so I cannot give any feedback on that part, but as an expert in the field of data science, I can say that the part describing the machine learning setup you are using is not detailed enough for anyone else to reproduce it or even judge whether your approach is sound.
As briefly described on original manuscript, we use Python system (line 142) and use Numpy utilities (https://numpy.org/) and Microsoft Excel macro.
May be this system is very elementally system for advanced computer scientist, however everyone who are not specialist in computer science could use it easily like us.
One of the appeals of this manuscript is that the simulation will supplement the training data, which could not be done experimentally, and will be meaningful to process engineers by providing a rough prediction of the results under conditions where the parameters have not been varied in the experiment.
Currently, we have no idea to open the source because computer scientist will provide more accurate and high-performance software systems.
I'll take you the time and effort but Please review again.
Thank you.
Yoshihiko HIRAI

Round 2
Reviewer 1 Report
I recommend to be published.